# Relationships between Sleep Quality, Introspective Accuracy, and Confidence Differ among People with Schizophrenia, Schizoaffective Disorder, and Bipolar Disorder with Psychotic Features

**DOI:** 10.3390/bs14030192

**Published:** 2024-02-28

**Authors:** Cassi R. Springfield, Amy E. Pinkham, Philip D. Harvey, Raeanne C. Moore, Robert A. Ackerman, Colin A. Depp, Kelsey A. Bonfils

**Affiliations:** 1School of Psychology, University of Southern Mississippi, Hattiesburg, MS 39406, USA; 2Department of Psychology, University of Texas at Dallas, Richardson, TX 75080, USA; 3Department of Psychiatry and Behavioral Sciences, Miller School of Medicine, University of Miami, Miami, FL 33136, USA; 4Bruce W. Carter VA Medical Center, Miami, FL 33125, USA; 5Department of Psychiatry, University of California San Deigo, San Diego, CA 92093, USA; 6San Diego VA Medical Center, San Diego, CA 92161, USA

**Keywords:** self-awareness, sleep disturbance, introspective bias, insight

## Abstract

People with schizophrenia-spectrum and bipolar disorders have difficulty accurately estimating their abilities and skills (impaired introspective accuracy [IA]) and tend to over- or underestimate their performance. This discrepancy between self-reported and objective task performance has been identified as a significant predictor of functional impairment. Yet, the factors driving this discrepancy are currently unclear. To date, the relationships between sleep quality and IA have not been examined. The current study aimed to explore the relationships between sleep quality and IA in participants diagnosed with schizophrenia (SCZ; *n* = 36), schizoaffective disorder (SCZ-A; *n* = 55), and bipolar disorder with psychotic features (BP; *n* = 87). Participants completed tasks of emotion recognition, estimated their performance on the tasks (used to calculate IA), and provided confidence ratings for their accuracy judgments. Participants also self-reported their sleep quality. These results suggest significantly greater discrepancies between self-reported and actual task scores for those with SCZ and SCZ-A compared to participants with BP. For those with SCZ, *lower* confidence on the tasks and *underestimation* of abilities were associated with lower sleep quality, while for those with SCZ-A, lower sleep quality was associated with *higher* confidence and *overestimation* of performance. Results suggest differential relationships between diagnostic groups. Future research is needed to further explore the factors driving these differing relationships, particularly the contrasting relationships between SCZ and SCZ-A.

## 1. Introduction

Poor self-awareness is an important component of serious mental illnesses, such as schizophrenia-spectrum and bipolar disorders, and includes limited insight regarding clinical symptoms, cognitive capacity, and functional abilities [1,2,3,4,5,6]. One self-evaluative process that has garnered increased attention in recent years is introspective accuracy (IA). IA refers to one’s ability to accurately estimate their own abilities, skills, and performance [7], and impairments in IA are indexed by differences between one’s self-report of their abilities and their actual performance in those domains. People with schizophrenia-spectrum and bipolar disorders demonstrate difficulty in accurately evaluating their neurocognitive, social cognitive, and functional abilities [2,8,9,10,11,12]. Deficits in IA are bidirectional, and people have been found to both under- and overestimate their abilities [2,13,14,15,16]. The direction of IA discrepancies (i.e., under- and overestimation) has been referred to as “introspective bias” (IB) [16]. In other words, IA is the accuracy of one’s self-assessment, while IB is the direction of one’s self-assessment errors [17]. Underestimation of abilities can be referred to as negative IB, while overestimation of abilities can be referred to as positive IB [17]. Both IA and IB have important implications for functional outcomes. Bidirectional misestimation of ability (i.e., both under- and overestimation) has been identified as more predictive of deficits in everyday functioning than actual performance on neurocognitive tests or measures of functional capacity [2,4], while overestimation of social cognitive abilities has been associated with poorer informant-rated social functioning, regardless of participants’ actual performance on assessments of social cognition [18].

Related to IA is confidence in task performance. Previous work suggests that people with schizophrenia-spectrum and bipolar disorders report confidence in their abilities that is unrelated to their objective performance. For example, one study identified that participants were overly confident while completing tasks of social cognition, regardless of their actual performance on the tasks, sometimes even reporting that their performance was perfect [13]. Additional work suggests moderate correlations between confidence ratings and item-by-item accuracy judgments, while confidence ratings were unrelated to actual task scores on a neurocognitive task [12]. Confidence about task performance appears to be related to both IA and IB, and in expected directions: higher confidence is associated with overestimation of abilities, and lower confidence is associated with underestimation of abilities [12,13]. These relationships can reflect one’s general tendency to either over- or underestimate their abilities [4,13,17].

The interrelations between IA, IB, and confidence are supported by research defining IA impairment as including item-by-item judgments (e.g., “Do you think you answered correctly”), confidence in those judgments, and the ability to combine judgments and feedback into global assessments of functioning or performance [19]. While IA seems broadly impaired in schizophrenia-spectrum and bipolar disorders, impairments may occur at different stages of this self-evaluation process between diagnostic groups [19]. Impairments can be observed at the item-by-item level, in making confidence judgments, or in incorporating feedback and previous judgments into global estimations of performance [19]. Given research suggesting greater severity of cognitive and social cognitive impairment and poorer functional outcomes in schizophrenia-spectrum disorders compared to bipolar disorders [20,21], impairments in IA may also be more severe for those with schizophrenia-spectrum disorders. For example, one study found that while participants in both groups overestimated their abilities when making momentary item-by-item performance judgments on the Wisconsin Card Sorting Test, only participants with bipolar disorder were able to produce summary IA ratings after completing the task that was related to their actual task scores, demonstrating more intact IA for this group [12]. Subsequent work on IA for the Wisconsin Card Sorting Test further supports differences in these groups [22]. While both groups were able to correctly identify when they answered incorrectly, participants with bipolar disorder were more able to successfully integrate feedback and their accuracy judgments into their task performance. In contrast, however, recent work directly comparing schizophrenia-spectrum and bipolar disorder groups suggests no differences in IA between diagnostic groups [19,23]. Specifically, in both studies, no differences in absolute value IA for the MATRICS Cognitive Consensus Battery were found between diagnostic groups [19,23]. The methodological differences in how IA is measured (i.e., summary IA ratings versus absolute value IA) and the domains of IA assessed in these studies may be contributing to this discrepancy. More work is needed to further explore these diagnostic differences, and critically, work examining those with schizophrenia and schizoaffective disorder separately is needed. To date, most of the work on IA combines people with schizophrenia and schizoaffective disorders into one schizophrenia-spectrum disorders group [12,17,23,24]; examining these diagnostic groups separately may explain mixed findings.

Research suggests that one’s tendency to over- or underestimate their abilities may be related to different determinants. For example, severe depressive symptoms have been associated with greater deficits in IA and underestimation of abilities (i.e., negative IB) [11]. In contrast, participants who report minimal levels of sad mood have a positive IB, demonstrated by overestimations of performance [25,26]. Overestimations of ability have also been associated with greater symptoms of psychosis in both cross-sectional [10] and momentary longitudinal assessments [27]. One study found that more auditory hallucinations, paranoia, and delusions across a 30-day period were related to greater impairments in IA for those with schizophrenia-spectrum disorders but not participants with bipolar disorder [19] whose momentary psychotic symptoms were too infrequent to be examined. Additional work points to executive functioning deficits as a possible determinant of IA impairment, specifically a failure to appropriately attend to and incorporate feedback [12,22]. However, despite this work, these relationships are complex, and additional efforts are needed to elucidate the mechanisms underlying IA impairment at the individual component level and to explore how these interrelations may vary between diagnostic groups. Given the breadth of IA impairments across populations and associations with functional outcomes, better understanding of the determinants of IA, IB, and confidence is imperative.

One possible determinant of IA that has not been previously explored is sleep quality. Sleep difficulties occur in up to 80% of people with schizophrenia-spectrum and bipolar disorders [28,29,30], and may include insomnia, hypersomnia, circadian rhythm abnormalities, obstructive sleep apnea, poor sleep quality, or other sleep disturbances [29,31,32]. Poor sleep quality in these populations is linked to more severe symptoms, greater suicide risk, impaired functioning, and lower quality of life [28,33,34,35]. The relationships between sleep difficulties and impaired cognitive processing across populations are also important to note [36,37,38], given research suggesting that higher-order cognitive abilities (e.g., executive functioning abilities) may be related to IA [12,39]. Additionally, poor sleep quality has been associated with greater distress in these groups and may impact how one is able to cope with stressors in daily life [40,41]. For example, in one study with participants with schizophrenia-spectrum disorders, greater sleep difficulties were associated with a general tendency to appraise stressors and daily challenges in a negative light, which was related to impaired functioning and lower quality of life [40]. As such, it is possible that those with poor sleep quality are more likely to display a negative IB (i.e., underestimation of their abilities and performance), or report lower confidence in their capabilities. However, the relationships between sleep, IA, IB, and confidence have yet to be investigated.

In sum, people with schizophrenia-spectrum and bipolar disorders have difficulty accurately evaluating their abilities and skills and show biases towards under- and overestimation of their performance on tasks. Research suggests that mood, symptoms, and cognitive abilities may be related to IA and may have specific implications for the direction of IB, but more work is needed to fully understand these complex relationships. The current study had two primary aims. First, we aimed to compare IA between participants with schizophrenia, schizoaffective disorder, and bipolar disorder with psychotic features. Second, we sought to examine the relationships between sleep quality, IA, IB, and confidence, and to investigate if these relationships differ between diagnostic groups. We expected that participants with schizophrenia and schizoaffective disorder would demonstrate greater impairments in IA compared to participants with bipolar disorder. We hypothesized that a greater discrepancy between self-reported and actual task scores (as indexed by absolute value IA) would be associated with poorer sleep quality. Additionally, we predicted that poorer sleep quality would be related to underestimations of ability (i.e., negative IB) and lower confidence about performance across groups. We did not have specific hypotheses regarding differences in the relationships between IA, IB, confidence, and sleep quality between the diagnostic groups; these analyses were considered exploratory.

## 2. Materials and Methods

### 2.1. Participants

To be eligible to participate, individuals were required to: (1) be between the ages of 18 and 65, (2) meet DSM-5 criteria for schizophrenia, schizoaffective disorder, or bipolar I with psychotic features, and (3) be clinically stable (i.e., no psychiatric hospitalizations or medication changes for at least six weeks prior to participation). To differentiate schizophrenia from schizoaffective disorder, participants must report significant mood episodes (i.e., depressive or manic episodes) that occur for the majority (i.e., greater than 50%) of the total duration of illness to be diagnosed with schizoaffective disorder [42]. Exclusion criteria included: (1) head trauma with loss of consciousness, (2) medical or neurological disorders, (3) neurodegenerative disorders, (4) significant visual or auditory impairments, (5) pervasive developmental disorders or intellectual disabilities (defined as a standard score of less than 70 on the Reading Subtest of the Wide-Range Achievement Test-4 [43]), or (6) Alcohol Use Disorder or Substance Use Disorder within the past three months.

Participants were recruited as part of a larger longitudinal study examining suicidality in psychotic disorders across three study sites in Dallas, TX, Miami, FL, and San Diego, CA, USA. Participants were recruited from outpatient mental health clinics, hospitals, and local community clinics, as well as from internet sources, such as Facebook, Craigslist, lab websites, and digital marketing companies. As part of the aims of the larger study, participants scoring a two or greater on suicidal ideation in the past month or who had suicidal behaviors in the past three months (as assessed by the Columbia Suicide Severity Rating Scale) were enrolled in the actively suicidal group. Individuals not meeting these criteria were enrolled in the non-suicidal group. To address the aims of the larger study, participants meeting criteria for the actively suicidal group were oversampled such that there were approximately equal numbers of participants in both groups in the final sample. For the purposes of the current analyses, participants were collapsed within diagnostic categories. The current study included 178 participants. Participant demographics can be seen in Table 1.

### 2.2. Measures

Introspective Accuracy (IA) and Introspective Bias (IB). Participants completed two widely used measures of emotion recognition: the Bell–Lysaker Emotion Recognition Task (BLERT) and the Penn Emotion Recognition Test (ER-40) [44,45]. To assess IA and IB, participants were asked “Do you think you answered correctly?” with a dichotomous yes–no answer choice immediately after each item of the BLERT and ER-40. Items that participants indicated they answered correctly received one point each, and self-reported scores on all items were summed together resulting in a total self-reported score that had the same range as the actual total score on the task (i.e., 0–21 on the BLERT; 0–40 on the ER-40). The actual task score was then subtracted from the participant’s self-reported total score. These difference scores were used as an index of IB (i.e., negative IB values indicate an underestimation of abilities and positive IB values indicate an overestimation of abilities) [2,12] which will be used to examine the direction of IA impairment. Difference scores were then turned into an absolute value, which created an absolute value IA [19,23]. A higher absolute value IA indicates a greater discrepancy between self-reported and objective task scores. IA and IB were examined separately for both tasks. Of note, there is not a separate measure used to assess IA/IB. The questions used to assess IA/IB (i.e., “Do you think you answered correctly?”) are embedded within the BLERT and ER-40 tasks and are presented to participants immediately after they select which emotion they think is being expressed in the presented stimuli.

Confidence. To examine confidence about performance, participants were asked to indicate how confident they were in the correctness of their choice for each item of the BLERT and ER-40. For the BLERT, participants indicated their confidence from 0 (not at all confident) to 100% (extremely confident) [18]. For the ER-40, participants indicated their confidence as 0% (not at all confident), 25%, 50%, 75%, or 100% (very confident) [39]. Confidence ratings on each item were averaged into a single average confidence score, which was calculated separately for each task [12,13].

Pittsburgh Sleep Quality Index (PSQI). The PSQI is a self-report measure of sleep quality over the past month [46]. The PSQI produces a global sleep quality score and seven component scores: subjective sleep quality, sleep disturbance, sleep latency, sleep duration, sleep efficiency, use of sleep medication, and daytime dysfunction. The PSQI includes 19 items, and higher scores indicate worse sleep quality (range: 0–21). The PSQI is a widely used measure of sleep quality among those with schizophrenia-spectrum and bipolar disorders [47].

Columbia Suicide Severity Rating Scale (C-SSRS). The C-SSRS assesses the severity of current and lifetime suicidal ideation and behavior [48]. Suicidality was assessed as a potential covariate in the current study.

### 2.3. Procedure

The current study uses data collected during the baseline visit of a year-long longitudinal study. The Mini International Neuropsychiatric Interview (MINI) [49] and the psychosis module of the Structured Clinical Interview for DSM-5—Research Version (SCID-5-RV) [50] were administered by trained raters to determine diagnoses. Raters were specifically trained on how to make differential diagnoses between schizophrenia, schizoaffective disorder, and bipolar disorder with psychotic features. The participants then completed the BLERT and ER-40 emotion recognition tasks, made IA and confidence ratings, and completed the PSQI self-report measure. Baseline visits occurred in person or remotely, based on COVID-19 safety guidelines and participant preference during the pandemic. All participants provided informed consent and were compensated for their time and participation. The study procedures were approved by the Institutional Review Boards at each study site.

### 2.4. Statistical Analysis

First, suicidality was evaluated as a potential covariate through a series of independent *t*-tests examining group differences between those enrolled in the actively suicidal group and the non-suicidal group for our variables of interest.

Group differences in absolute value IA, objective task scores, and confidence ratings between the diagnostic groups were examined using ANCOVAs, and significant differences were followed up with Bonferroni post hoc comparisons. One-sample *t*-tests were used to determine if the absolute value IA significantly differed from zero, indicating significant impairment in IA.

To examine the associations between IA, IB, confidence, objective task scores, and sleep quality, subscale scores from the PSQI were correlated with the absolute IA values, IB difference scores, average confidence ratings, and total task scores using partial correlations to control for relevant covariate(s). These analyses were conducted in the diagnostic groups separately. Of note, given the early nature of this line of research, a traditional significance value of 0.05 was used in evaluating correlational relationships and we did not correct for multiple comparisons. The effect sizes of the relationships were considered in understanding their significance. To directly compare the magnitude of significant correlations between the diagnostic groups and account for the variation in sample sizes between the groups, Fisher’s r-to-z transformations were used.

## 3. Results

The mean sleep quality scores for the actively suicidal and the non-suicidal groups, as well as comparisons between the groups, can be seen in Appendix A. The actively suicidal group had marginally poorer overall sleep quality than the non-suicidal group. There were also significant differences between groups on several PSQI subscales (e.g., subjective sleep quality, sleep disturbances). Given these group differences, participants’ group status (i.e., enrolled in the actively suicidal vs. non-suicidal group) was included as a covariate in all analyses.

Mean sleep quality, task scores, IA absolute values, and IB difference scores in the diagnostic groups can be seen in Table 2. The diagnostic groups differed on IA absolute values on both the BLERT (*F*(2, 158) = 6.22, *p* = 0.003, partial eta-squared = 0.07) and ER-40 tasks (*F*(2, 163) = 3.19, *p* = 0.04, partial eta-squared = 0.04). On the BLERT, the schizoaffective disorder group demonstrated a significantly larger discrepancy between self-reported and actual scores (*Estimated mean* = 6.10, *SE* = 0.50) than the bipolar disorder group (*Estimated mean* = 4.03, *SE* = 0.40; *p* = 0.004). Additionally, the schizophrenia group (*Estimated mean* = 5.83, *SE* = 0.64) demonstrated a greater discrepancy between self-reported and actual scores on the BLERT than the bipolar disorder group (*p* = 0.05), which was trending in significance. Similarly, on the ER-40 task, the schizophrenia group (*Estimated mean* = 10.11, *SE* = 0.99) had a significantly greater discrepancy between self-reported and objective scores compared to the bipolar group (*Estimated mean* = 7.12, *SE* = 0.65; *p* = 0.04); no other differences between diagnostic groups emerged. There were no differences between the diagnostic groups in confidence ratings for either task (BLERT: (*F*(2, 158) = 0.13, *p* = 0.88, partial eta-squared = 0.002); ER-40: (*F*(2, 163) = 0.35, *p* = 0.71, partial eta-squared = 0.004). When examining objective task scores, groups differed on actual task scores for the BLERT (*F*(2, 158) = 7.69, *p* < 0.001, partial eta-squared = 0.09). Specifically, the bipolar disorder group (*Estimated mean* = 16.30, *SE* = 0.40) earned higher scores on the BLERT than the schizophrenia (*Estimated mean* = 14.04, *SE* = 0.65; *p* = 0.01) and schizoaffective disorder groups (*Estimated mean* = 14.10, *SE* = 0.50; *p* = 0.002). There were no differences in ER-40 task scores between the diagnostic groups (*F*(2, 163) = 1.43, *p* = 0.24, partial eta-squared = 0.02).

One-sample *t*-tests within each group comparing absolute value IA to 0 indicated impairments in IA for those with bipolar disorders for the ER-40 (*t*(77) = 11.45, *p* < 0.001, *d* = 1.30) and BLERT (*t*(79) = 12.68, *p* < 0.001, *d* = 1.42), indicating that the difference between self-reported scores and objective performance was statistically significant. Similarly, the schizophrenia group also demonstrated a significant discrepancy between self-reported scores and actual performance on both tasks (ER-40: *t*(33) = 8.15, *p* < 0.001, *d* = 1.40; BLERT: *t*(30) = 8.14, *p* < 0.001, *d* = 1.46), as did the schizoaffective group (ER-40: *t*(54) = 11.69, *p* < 0.001, *d* = 1.58; BLERT: *t*(50) = 10.40, *p* < 0.001, *d* = 1.46).

Partial correlations (including suicidality as a covariate) were conducted to examine relationships between sleep quality and absolute value IA, IB difference scores, confidence ratings, and objective task scores in the diagnostic groups separately (Table 3). In the bipolar group, lower confidence on the ER-40 was associated with longer sleep duration and using sleep medication more frequently. Similarly, relationships with IB difference scores indicated that underestimation of ER-40 performance (i.e., self-reported scores were lower than actual ER-40 task scores) was associated with longer sleep duration and greater use of sleep medication in this group. In the schizophrenia group, lower confidence on the BLERT was associated with longer sleep latency, shorter sleep duration, poorer sleep efficiency, more sleep disturbances, and poorer overall sleep quality. Similarly, lower confidence on the ER-40 in this group was associated with greater daytime dysfunction, and underestimation of ER-40 performance was associated with poorer sleep quality, longer sleep latency, shorter sleep duration, greater sleep disturbances, and overall poorer sleep. In contrast, in the schizoaffective disorder group, the directions of these relationships differed: higher confidence on both tasks was associated with more sleep disturbance and overestimation of ER-40 performance (i.e., self-reported scores were higher than actual ER-40 task scores) was associated with longer sleep latency. No significant relationships were revealed between sleep quality and absolute value IA, or between sleep quality and objective task scores on the BLERT and ER-40.

Given that significant relationships between some variables emerged in more than one diagnostic group, and the different sample sizes between diagnostic groups, Fisher’s r-to-z transformations were used to directly compare the strengths of these correlations. Relationships between sleep disturbance and confidence on the BLERT were stronger for those with schizoaffective disorder compared to those with schizophrenia (*z* = −3.39, *p* < 0.001), as were the relationships between sleep latency and ER-40 IB (*z* = −3.32, *p* < 0.001). When comparing relationships between sleep duration and ER-40 IB between the bipolar disorder and schizophrenia groups, relationships were stronger for those with bipolar disorder (*z* = 2.95, *p* = 0.003).

Partial correlations including suicidality as a covariate were conducted to examine relationships between IB difference scores and confidence ratings in the diagnostic groups. IB difference scores were significantly correlated with confidence ratings in the bipolar group (ER-40: *r* = 0.52, *p* < 0.001; BLERT: *r* = 0.24, *p* = 0.04), suggesting that higher confidence was associated with overestimation of performance. The direction of these relationships was consistent in the schizophrenia and schizoaffective disorder groups (i.e., higher confidence associated with overestimation), though the relationships varied between the tasks. In the schizophrenia group, only BLERT confidence and BLERT IB were significantly correlated (*r* = 0.47, *p* = 0.008). In the schizoaffective disorder group, relationships were only significant for ER-40 confidence and ER-40 IB (*r* = 0.38, *p* = 0.005).

## 4. Discussion

Broadly, the results from this study are consistent with a growing body of work highlighting impairments in IA across schizophrenia-spectrum and bipolar disorders [2,12,13,17,19,23]. Results from the current study indicate that, although all diagnostic groups demonstrated impaired IA, those with schizophrenia-spectrum disorders displayed larger misestimations in performance than those with bipolar disorder. In other words, people with schizophrenia-spectrum disorders displayed a greater discrepancy between their self-reported and actual scores on the tasks. This is consistent with some prior work indicating that while both diagnostic groups tend to overestimate their abilities when making item-by-item judgments, participants with bipolar disorder demonstrate more intact IA when asked to produce summary IA ratings after completing a task [12], and they are better able to process and incorporate feedback into their performance [22]. However, this is inconsistent with recent work suggesting no differences in absolute value IA between diagnostic groups [19,23]. Differences in how IA is measured and analyzed in these studies, as well as the tasks used to generate IA ratings, may explain these discrepancies in part, though future work should continue to investigate these mixed findings. Despite differences between absolute value IA, confidence in performance was not different between the diagnostic groups. This further supports research suggesting that IA involves multiple interrelated but distinct components and that impairments in IA may occur at different stages of this process, which may vary between schizophrenia-spectrum and bipolar disorders [19].

This study is the first to investigate the relationships between sleep quality, IA, IB, and confidence. Contrary to expectations, we did not observe significant relationships between sleep quality and absolute value IA; however, interesting relationships emerged between specific aspects of sleep quality, IB, and confidence ratings on the tasks. For those with bipolar disorder, greater frequency of sleep medication use (considered poorer sleep quality on the PSQI) was associated with both underestimation of performance and lower confidence. Lower confidence and underestimation of performance were also associated with longer sleep duration in this group. In the schizophrenia group, underestimation of performance and lower confidence were consistently associated with greater sleep disturbance across multiple domains (i.e., poorer subjective sleep quality, longer sleep latency, shorter sleep duration, poorer sleep efficiency, more sleep disturbances, greater daytime dysfunction, and poorer overall sleep quality). Again, opposite patterns emerged for those with schizoaffective disorder: overestimation of performance was associated with longer sleep latency, and higher confidence was associated with more sleep disturbances. While similar relationships were indicated between diagnostic groups, Fisher’s r-to-z transformations suggest that the magnitude of these relationships differed. Specifically, correlations between sleep disturbance and confidence on the BLERT, as well as correlations between sleep latency and ER-40 IB, were stronger for those with schizoaffective disorder compared to those with schizophrenia. Additionally, correlations between sleep duration and ER-40 IB were stronger for those with bipolar disorder compared to those with schizophrenia. Of note, no significant relationships were revealed between objective task scores and sleep quality, suggesting that the observed relationships between IB, confidence, and sleep quality in the current study cannot be attributed to the impact of sleep on general task performance. Associations between sleep quality, IB, and confidence in the current study are novel and indicate that sleep quality may be a determinant of both IB and confidence, which may have unique relationships between diagnostic groups.

Our finding that sleep quality was associated with IB and confidence, but not absolute value IA, is deserving of discussion, as these constructs were developed in tandem and previous research suggests moderate correlations between IA and confidence [12]. One possible explanation is that sleep quality has specific implications for one’s tendency to over- or underestimate their abilities and skills (i.e., IB and confidence), rather than misestimating performance in general (i.e., absolute value IA). In other words, it may be the direction of these impairments that is important to consider when examining relationships with sleep. Alternatively, another possible explanation is that these differing associations are related to methodological differences in how these constructs are assessed. Specifically, the nature of the confidence questions allows for a wider range of responses (i.e., 0 to 100%) than the dichotomous IA questions, and it is possible that confidence ratings may capture greater variability and fluctuations in assessments of one’s own abilities and skills between items on a task. Similarly, the absolute IA value does not take into account the direction of the discrepancy between self-reported and actual task scores (i.e., under- or overestimation), whereas both the IB and confidence rating values are bidirectional, which also increases variability. Use of the absolute IA value may reduce statistical sensitivity such that these nuanced relationships are not detected. Ultimately, research examining IA is relatively new, and consensus regarding definitions and measurement methods in the field is needed. As the research examining IA continues to grow, investigators should continue to explore different ways to assess the different components of IA and how they are interrelated.

Differential relationships between diagnostic groups are of interest and warrant further discussion, particularly the contrasting relationships between the schizophrenia and schizoaffective disorder groups. One possible explanation for these findings is the greater symptom complexity associated with schizoaffective disorder, specifically the significant mood episodes that distinguish schizoaffective disorder from schizophrenia [42]. This mood component may drive the differential relationships observed in the current study. Indeed, mood has often been considered integral to understanding how one perceives their abilities and skills [7,16], and the exact relationships between mood, IA, IB, and confidence are still being investigated. To date, a large majority of the research in the IA literature combines people with schizophrenia and schizoaffective disorders into one group (i.e., schizophrenia-spectrum disorders) [12,17,23,24]. It should also be noted that differentiating the diagnoses of schizophrenia and schizoaffective disorder is often difficult, and some work suggests low test–retest reliability of schizoaffective disorder (for a meta-analysis see [51]). This may be related to participants’ challenges in self-assessments of their emotional and mood states. For example, momentary sad mood has been found to be underreported in schizophrenia [26], while some participants will report that they are experiencing both positive and negative mood simultaneously, contrary to the normative inverse relationships between positive and negative mood states [52]. Nonetheless, given the findings from the current study, it may be especially important to consider the diagnostic differences in research examining IA, IB, and confidence. Exploring these relationships in diagnostic groups independently, specifically separating schizophrenia-spectrum disorders, may help to better elucidate specific determinants of IA, IB, and confidence.

Of note, we did observe some differences in our findings between the two social cognitive tasks used to generate IA, IB, and confidence ratings. While both tasks assess emotion recognition broadly, different skills are required in each task. For example, the BLERT involves more complex skills, as participants watch videos incorporating facial, vocal, and upper-body movement cues to determine the expressed emotion [44]. In contrast, the ER-40 tests facial affect recognition from static images [45]. In the current study, links between IB and sleep were present only for the ER-40, while relationships between confidence and sleep were documented for both tasks. This suggests that sleep may be important for understanding IB when completing a facial emotion recognition task but may not be as important for multi-modal emotion recognition. In the more complex BLERT task, participants have access to many different sources of information including facial cues, vocal sounds, and upper-body movements. Perhaps sleep is less impactful on IB because participants have more sources of information available to use when monitoring their performance. Conversely, for the simplified ER-40 task, the limited stimuli may allow sleep to have a stronger impact on their momentary judgments about performance because there is little information available for participants to use when making these determinations. Future research should further investigate these questions.

While this study has several strengths, its limitations must also be considered. First, the schizophrenia group was smaller than the other two diagnostic groups in this study, which may have impacted our between-group analyses. Findings from this study should be replicated in larger samples. Additionally, 80% of participants completed this study remotely due to COVID-19 safety precautions and participant preference during the pandemic. The differences in study methods between remote and in-person participation (as well as the impact of the pandemic broadly on mental health) are important to consider when interpreting these findings. Future investigations should explore the relationships between IB, confidence, and sleep using in-person assessments. Given the novelty of this line of work, we elected to use a traditional significance value of 0.05 for our correlational analyses and did not correct for multiple comparisons. Future work should attempt to replicate some of the observed correlational relationships, particularly those with smaller effect sizes, in additional samples. Lastly, this study relied on retrospective self-reports of sleep quality. Previous work suggests low correlations between self-reported and behavioral assessments of sleep, with participants often overreporting sleep duration on self-reports as compared to objective measures [53]. The interrelations between sleep quality, IA, IB, and confidence should be examined using objective indicators of sleep quality, such as actigraphy.

Results from this study suggest greater IA impairments among those with schizophrenia-spectrum disorders compared to bipolar disorders; however, diagnostic groups did not differ in their confidence in their abilities. Results also indicate unique relationships between sleep quality, confidence, and over- and underestimations of performance (i.e., IB) between diagnostic groups. Future work should incorporate ambulatory assessments of sleep to explore how these relationships may compare to self-reports of sleep quality. Additionally, future investigations should also assess how symptoms and mood states may impact these relationships. Given the link between greater symptoms and poorer sleep quality in these groups [28,33,34,35], it is possible that the observed relationships in the current study may be partially attributed to the impacts of elevated symptomology on IA. Longitudinal methods exploring relationships between day-to-day sleep, symptoms, IA, IB, and confidence may aid in further defining these relationships, particularly in our understanding of their temporal nature.

## Figures and Tables

**Table 1 behavsci-14-00192-t001:** Sample demographic and descriptive characteristics.

	*n* (%)/*M* (*SD*)
Diagnoses	
Bipolar Disorder with psychotic features	87 (48.9%)
Schizophrenia	36 (20.2%)
Schizoaffective Disorder	55 (30.9%)
Age	40.28 (11.61)
Years of education	14.17 (2.75)
Race	
American Indian or Alaskan Native	1 (0.6)
Asian	5 (2.8)
Black or African American	57 (32.0)
White	84 (47.2)
Native Hawaiian or Other Pacific Islander	2 (1.1)
More than one race or other	29 (16.3)
Ethnicity	
Hispanic/Latinx	41 (23.0)
Non-Hispanic/Latinx	137 (77.0)
Gender identity	
Male	56 (31.5)
Female	121 (68.0)
Non-binary	1 (0.6)
Suicidality group	
Actively suicidal	83 (46.6)
Non-suicidal	95 (53.4)

**Table 2 behavsci-14-00192-t002:** Average sleep quality and task scores.

	Bipolar Disorder	Schizophrenia	Schizoaffective Disorder
	*M* (*SD*)	*M* (*SD*)	*M* (*SD*)
PSQI Total	9.92 (4.16)	8.22 (4.03)	9.33 (4.09)
PSQI Component 1	1.51 (0.86)	1.03 (1.00)	1.15 (0.99)
PSQI Component 2	1.84 (1.04)	1.53 (1.00)	1.56 (1.08)
PSQI Component 3	1.06 (1.06)	0.75 (1.16)	0.84 (1.10)
PSQI Component 4	1.08 (1.09)	1.00 (1.24)	0.80 (1.08)
PSQI Component 5	1.63 (0.67)	1.22 (0.76)	1.58 (0.71)
PSQI Component 6	1.30 (1.37)	1.22 (1.33)	1.47 (1.39)
PSQI Component 7	1.46 (0.89)	1.03 (1.03)	1.29 (0.98)
ER-40 Scores			
ER-40 Objective Score	29.41 (7.39)	26.94 (7.74)	28.11 (8.11)
ER-40 Self-reported Score	33.04 (11.62)	34.26 (11.22)	32.69 (12.72)
ER-40 IA Absolute Value	7.12 (5.49)	10.03 (7.18)	8.18 (5.19)
ER-40 IB Difference Score	3.63 (8.25)	7.32 (10.00)	4.58 (8.59)
ER-40 Confidence Ratings	78.33 (14.84)	80.59 (16.39)	77.56 (14.99)
BLERT Scores			
BLERT Objective Score	16.28 (2.87)	14.07 (3.97)	14.08 (4.27)
BLERT Self-reported Score	19.91 (1.93)	19.23 (3.30)	19.86 (2.04)
BLERT IA Absolute Value	4.04 (2.85)	5.81 (3.97)	6.10 (4.19)
BLERT IB Difference Score	3.64 (3.35)	5.16 (4.80)	5.78 (4.62)
BLERT Confidence Ratings	82.11 (14.85)	81.17 (13.90)	80.85 (16.14)

Note: PSQI Total = Global PSQI Score; PSQI Component 1 = Subjective sleep quality; PSQI Component 2 = Sleep latency; PSQI Component 3 = Sleep duration; PSQI Component 4 = Sleep efficiency; PSQI Component 5 = Sleep disturbances; PSQI Component 6 = Use of sleeping medication; PSQI Component 7 = Daytime dysfunction; ER-40 IA Absolute Value = Absolute value of self-reported score − actual score on the ER-40; BLERT IA Absolute Value = Absolute value of self-reported score − actual score on the BLERT; ER-40 IB Difference Score = Self-reported score − objective score on the ER-40; BLERT IB Difference Score = Self-reported score − objective score on the BLERT.

**Table 3 behavsci-14-00192-t003:** Partial correlations between confidence ratings, absolute value IA, IB difference scores, task scores, and sleep quality.

	PSQI 1	PSQI 2	PSQI 3	PSQI 4	PSQI 5	PSQI 6	PSQI 7	PSQI Total
**Bipolar Disorder**								
BLERT Confidence	0.06	−0.16	0.15	−0.09	−0.05	−0.09	−0.09	−0.07
ER-40 Confidence	0.01	−0.17	**0.28 ***	−0.01	0.04	**−0.25 ***	−0.11	−0.08
BLERT IA	−0.01	0.19	0.05	0.09	0.16	0.12	0.05	0.14
ER-40 IA	−0.06	0.12	−0.04	0.003	0.04	0.22 ^†^	−0.11	0.06
BLERT IB	−0.02	0.07	0.02	−0.06	0.14	0.03	0.08	0.04
ER-40 IB	−0.06	−0.09	**0.25 ***	0.04	0.08	**−0.26 ***	−0.09	−0.05
BLERT Objective Score	0.12	−0.15	0.12	0.03	−0.07	−0.05	−0.11	−0.01
ER-40 Objective Score	0.10	−0.02	0.17	−0.04	−0.02	−0.11	0.01	0.01
**Schizophrenia**								
BLERT Confidence	−0.13	**−0.42 ***	**−0.43 ***	**−0.44 ****	**−0.46 ****	−0.28	−0.23	**−0.48 ****
ER-40 Confidence	−0.20	−0.32	−0.12	−0.03	−0.24	−0.14	**−0.37 ***	−0.25
BLERT IA	−0.13	−0.21	0.04	−0.18	−0.03	0.29	0.03	0.13
ER-40 IA	−0.11	−0.08	−0.06	−0.03	−0.15	0.23	−0.05	0.08
BLERT IB	−0.10	−0.23	−0.10	−0.29	−0.08	0.21	0.04	−0.003
ER-40 IB	**−0.45 ****	**−0.37 ***	**−0.36 ***	−0.28	**−0.37 ***	0.01	−0.06	**−0.38 ***
BLERT Objective Score	0.14	0.17	−0.05	0.14	−0.02	−0.30	0.003	−0.12
ER-40 Objective Score	0.003	−0.10	−0.07	−0.02	0.03	−0.16	0.10	−0.19
**Schizoaffective Disorder**								
BLERT Confidence	0.09	−0.15	0.05	−0.01	**0.30 ***	−0.11	−0.002	−0.09
ER-40 Confidence	0.12	−0.15	0.08	0.04	**0.33 ****	−0.21	0.01	−0.01
BLERT IA	0.05	0.11	0.13	0.13	0.15	0.20	0.11	0.03
ER-40 IA	−0.22	0.08	−0.07	0.06	−0.06	−0.08	0.10	−0.09
BLERT IB	0.08	0.12	0.12	0.12	0.15	0.22	0.07	0.07
ER-40 IB	0.03	**0.35 ***	0.15	0.10	0.12	0.03	0.08	0.07
BLERT Objective Score	0.07	0.00	−0.05	−0.04	0.02	−0.18	−0.11	−0.10
ER-40 Objective Score	0.05	−0.12	−0.08	−0.16	0.14	−0.10	−0.06	0.04

Note: PSQI 1 = Subjective sleep quality; PSQI 2 = Sleep latency; PSQI 3 = Sleep duration; PSQI 4 = Sleep efficiency; PSQI 5 = Sleep disturbances; PSQI 6 = Use of sleeping medication; PSQI 7 = Daytime dysfunction; PSQI Total = Global PSQI Score; BLERT Confidence = Average confidence ratings on the BLERT; ER-40 Confidence = Average confidence ratings on the ER-40; BLERT IA = Absolute value of self-reported score − actual score on the BLERT; ER-40 IA = Absolute value of self-reported score − actual score on the ER-40; BLERT IB = Self-reported score − actual score on the BLERT; ER-40 IB = Self-reported score − actual score on the ER-40. ** *p* < 0.01; * *p* < 0.05; ^†^ *p* < 0.06. Significant correlations are bolded.

## Data Availability

The data presented in this study are openly available in the National Institute of Mental Health (NIMH) Data Archive.

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
