# Peer review of "Relationships between Sleep Quality, Introspective Accuracy, and Confidence Differ among People with Schizophrenia, Schizoaffective Disorder, and Bipolar Disorder with Psychotic Features"

_behavsci, 2024, doi:10.3390/bs14030192_

Round 1

Reviewer 1 Report

Comments and Suggestions for Authors

The manuscript is focusing very interesting topic and covered by literature data, and provide new insight into the relationships between sleep quality, introspective accuracy, and confidence in patients with schizophrenia, schizoaffective disorder, and bipolar disorder with psychotic features. Despite the differences in the results of previously published data this manuscript offers a good basis for further investigations related to this issue. The manuscript is well written and organized, and deserves attention.

Reviewer 2 Report

Comments and Suggestions for Authors

Clarification of Findings:

The text mentions "greater misestimations in performance" and "discrepancy between self-reported and actual scores." It would be helpful to explicitly define what is meant by "misestimations" and how the discrepancy is calculated or measured.

Scientific Precision:

Ensure precision in language. For instance, the phrase "greater misestimations in performance" might be more scientifically rigorous if rephrased as "larger discrepancies between self-perceived and objectively measured task performance."

Consistency in Terminology:

Use consistent terminology throughout the text. For example, in some instances, "IA" is used to represent "Introspective Accuracy," while in others, "IA" stands for "Introspective Ability." It would be beneficial to maintain consistent abbreviations to avoid confusion.

Data Presentation:

When presenting results, consider providing effect sizes along with statistical significance. This will offer a more comprehensive view of the findings.

Discussion Section:

While discussing the implications of the study, consider exploring potential confounding variables or alternative explanations for the observed associations between sleep quality and introspective abilities.

Methodology Clarification:

Specify how introspective abilities (IA and IB) are measured. A brief explanation or reference to the instruments used would enhance the scientific clarity of the text.
